# Methylation Analysis of Urinary Sample in Non-Muscle-Invasive Bladder Carcinoma: Frequency and Management of Invalid Result

**DOI:** 10.3390/biomedicines11123288

**Published:** 2023-12-12

**Authors:** Francesco Pierconti, E. D. Rossi, V. Fiorentino, A. Bakacs, A. Carlino, E. Navarra, E. Sacco, A. Totaro, G. Palermo, L. M. Larocca, M. Martini

**Affiliations:** 1Institute of Pathology, Catholic University of Rome, Fondazione Policlinico Gemelli Roma, 00153 Rome, Italy; esther.rossi@policlinicogemelli.it (E.D.R.); arianna.bakacs@policlinicogemelli.it (A.B.); elena.navarra@policlinicogemelli.it (E.N.); 2Institute of Pathology, University of Messina, 98125 Messina, Italy; vincenzo.fiorentino@unime.it (V.F.); maurizio.martini@unime.it (M.M.); 3Institute of Pathology, UniCamillus Rome, 00131 Rome, Italy; angela.carlino@unicamillus.org (A.C.); luigimaria.larocca@unicamillus.org (L.M.L.); 4Institute of Urology, Catholic University of Rome, Fondazione Policlinico Gemelli Roma, 00168 Rome, Italy; emilio.sacco@policlinicogemelli.it (E.S.); angelo.totaro@policlinicogemelli.it (A.T.); giuseppe.palermo@policlinicogemelli.it (G.P.)

**Keywords:** bladder carcinoma, methylation analysis, urinary cytology, non-muscle-invasive carcinoma

## Abstract

Background: Numerous studies showed that methylation analysis represents a newly developed urinary marker based on DNA methylation changes in a panel of genomic biomarkers and it could represent a valid tool in terms of the diagnosis and prediction of high-grade urothelial carcinoma recurrences. One of the limits of the use of this new molecular method during a follow-up is represented by the number of invalid tests in routine practice. Method: A total of 782 patients with a diagnosis of non-muscle-invasive high-grade carcinoma (NMIBC) was studied. The Bladder EpiCheck test (BE) was performed together with cytology in all cases within 1 year after the end of treatment. In 402 patients, the urinary samples were voided urine (UV), while, in 380 cases, the samples were collected after bladder washing (IU). For all the patients with invalid BE results, a second BE test was performed following the instructions for use that indicated the test should be repeated with a new urinary sample in the case of an invalid result. Results: Analyzing the two different groups (UV and IU), we found the invalid BE results seemed to be not related to urinary samples (*p* = 0.13 Fisher’s exact test), suggesting that the collection method was not relevant in order to reduce the number of invalid tests. Conclusions: In the follow-up for NMIBC, for patients for whom a BE test is planned, a combined approach of cytology and a methylation test is recommended in order to repeat the BE test with an invalid result only in those cases with a cytological diagnosis of atypical urothelial cells (AUC) suspicious for high-grade urothelial carcinoma (SHGUC) and high-grade urothelial carcinoma (HGUC).

## 1. Introduction

With an incidence rate of almost half a million per year, bladder cancer (BC) is the ninth most common cancer worldwide. BC is categorized into muscle-invasive bladder cancer (MIBC) and non-muscle-invasive bladder cancer (NMIBC). The management of non-muscle-invasive bladder carcinoma (NMIBC) after transurethral resection of a bladder tumor consists of surveillance and intravesical therapy [1]. The initial intermediate treatment involves transurethral resection of the bladder tumor (TURBT) followed by intravesical Bacillus Calmette–Guérin (BCG), which has been the standard of care for decades. The intravesical BCG therapy includes an induction course of six weekly instillations and a maintenance course every 3–6 months for 1–3 years. This treatment is superior to TURBT alone in terms of recurrence and progression prevention in NMIBC patients.

The management of non-muscle-invasive bladder carcinoma (NMIBC) after transurethral resection of a bladder tumor consists of surveillance and intravesical therapy [1].

Cytology with cystoscopy represents the most efficient method currently available for the diagnosis of recurrent urothelial carcinoma during follow-up. In addition, numerous recent studies showed also that the Bladder EpiCheck test (Nucleix Ltd., San Diego, CA, USA) (BE), as newly developed urinary markers based on DNA methylation changes in a panel of genomic biomarkers, could represent a valid tool in terms of the diagnosis and prediction of high-grade urothelial carcinoma recurrences. In fact, this test showed a specificity of 88% and, excluding low-grade carcinoma, the sensitivity was 91.7, with an NPV of 99.3 [2,3,4,5,6,7,8,9,10].

One of the limits of the use of this new molecular method during follow-up is represented by the number of invalid tests in routine practice, which could be due to multiple reasons: correct specimen collection, transportation and storage of the urinary sample, and a limited number of urothelial cells. The suggestion for most of these cases is to collect a new urine sample from the patient with the repetition of the analysis, with higher costs and an increase in the response time.

The primary objective of the current study was to analyze, for the first time to our knowledge, the percentage of invalid BE tests in patients with high-grade urothelial carcinoma during follow-up, investigating if the specimen collection method of voided urine (UV) or bladder washing-instrumented urine (IU) may affect the number of invalid results. Moreover, we analyzed if the combination of cytology and BE test could reduce the number of second BE tests in cases with invalid BE results. In fact, retesting samples in the cases of invalid results could represent not only an expensive cost but also a source of wasted time.

## 2. Materials and Methods

### 2.1. Case Selection and Study Cohort

A total of 782 patients with a diagnosis of non-muscle-invasive high-grade carcinoma (NMIBC), admitted to our department from January 2018 to November 2021, were treated and followed for 1 year. The mean age of the patients was 72.5 years (age range 45–93 years) with the number of males at 485 while the women were 297. In all cases, formalin-fixed and paraffin-embedded sections were made [11]; in 596 cases, a diagnosis of high-grade papillary carcinoma was made. Considering the WHO 1973 and the TNM classification (2017), 360 were G3T1 and 236 were G2T1, while in 186 cases, a diagnosis of in situ carcinoma (CIS) was made (Table 1, according to WHO (1973) and TNM classification 2017) [12,13].

The treatment was an intravesical therapy of Bacillus Calmette–Guerin (BCG) in 586 patients, while 196 patients were treated with mitomycin C. During the follow-up, the patients were evaluated by voided urine cytology and white-light cystoscopy, according to the European Association of Urology Guidelines [1].

The Bladder EpiCheck test was performed together with cytology in all cases, within 1 year after the end of treatment. In 402 patients, the urinary samples were voided urine, while in 380 cases, the samples were collected after bladder washing.

For the cytological diagnosis, the slides were reviewed by two expert uro-cytologists (F.P. and M.M.); for cases in which a consensus among the cytologists could not be agreed upon, a third uro-cytologist (E.D.R.) was consulted to reach a group consensus [14].

In cases with cytological diagnoses of HGUC, SHGUC, or AUC, a cystoscopy was performed within 3 months after urinary cytology samples and bladder biopsies with resection of the neoplastic area or with multiple bladder biopsies in a random fashion were obtained during cystoscopy, and formalin-fixed and paraffin-embedded tissues were processed according to previous studies [15,16].

The histological criteria for the diagnosis of high-grade urothelial carcinoma are architectural and nucleocytoplasmic heterogeneity with mitosis and necrosis. Nuclear polymorphisms, granular or coarse chromatin, and prominent nucleoli are common features in malignant proliferation.

All patients with a cytological diagnosis as negative for urothelial carcinoma (NHGUC) were followed by repeated urine cytology, either voided specimens or bladder washing obtained at the follow-up cystoscopy.

For all the patients with invalid BE results, a second BE test was performed following the instructions for use that indicated the test should be repeated with a new urinary sample in the case of an invalid result. The method for collection of the urinary samples was the same as that used for the first molecular analysis.

All patient data were collected anonymously, and written, informed consent, as part of the routine diagnosis and treatment procedures, was obtained from patients or their guardians according to the Declaration of Helsinki; the study adhered to Good Clinical Practice guidelines.

### 2.2. Urine Cytology Processing Method

#### 2.2.1. Cytology

The samples were centrifugated for 10 min at 2000 revolutions per minute. The resulting pellets were resuspended in Thin Prep PreservCyt solution and were processed using the TP 5000 System (Hologic Inc., Rome, Italy) [17].

Cytological evaluation was performed using the Papanicolaou staining procedure and the diagnosis was formulated according to the Paris System for Reporting Urinary Cytology, classifying the cytological specimens as negative for high-grade urothelial carcinoma (NHGUC), atypical urothelial cells (AUC), suspicious for high-grade urothelial carcinoma (SHGUC), positive for high-grade urothelial carcinoma (HGUC), and unsatisfactory/nondiagnostic [18,19]. The presence of an increased N/C ratio, hyperchromasia, irregular nuclear membrane, and coarse chromatin represents the criteria for malignancy and a diagnosis of high-grade urothelial carcinoma (HGUC) or suspicious high-grade urothelial carcinoma (SHGUC) if these nuclear features are present in a few cells.

#### 2.2.2. Bladder EpiCheck Test

For the Bladder EpiCheck test (Nucleix Ltd.), the urine sample was centrifugated twice at 1000× *g* for 10 min at room temperature. DNA was extracted using the Bladder EpiCheck DNA extraction kit and was digested using a methylation-sensitive restriction enzyme, which cleaves DNA at its recognition sequence if it is unmethylated. The samples were prepared for the PCR assay using the Bladder EpiCheck test kit, and the results were analyzed using the Bladder EpiCheck software, Version 1.9. For the sample that passed the internal control validation, an EpiScore (a number between 0 and 100) was calculated; an EpiScore > or = to 60 indicates a positive result (high risk for HGUC), while a score <60 indicates a negative result (low risk for HGUC). Specifically in the group of positive results, an EpiScore > or = to 90 indicates HGUC [2].

### 2.3. Statistical Analysis

Statistical analysis was performed using MedCalc version 12.3.0 (MedCalc Software, Mariakerke, Belgium) and GraphPad-Prism 5 software (Graph Pad Software, San Diego, CA, USA). A comparison of categorical variables was performed by the chi-square test or the Fisher’s exact test, as appropriate. A *p*-value less than 0.05 was considered as statistically significant [20,21,22].

## 3. Results

In 58 out of 402 patients (14%) with urine voided (UV) samples and in 41 out of 380 cases (11%) with instrumented urine samples (IU), the BE test did not provide a valid result, suggesting that we should collect new urine samples from the patients. Analyzing the two different groups, we found the invalid BE results seemed to be not related to urinary samples (*p* = 0.13 Fisher’s exact test), suggesting that the collection method was not relevant in reducing the number of invalid tests (Table 2).

Moreover, in the group of UV samples with an invalid BE test, in 45 out of 58 cases (77.6%), a cytological diagnosis of NHGUC was made; in the remaining 13 cases, 6 cases showed a cytological diagnosis of HGUC, and in 5 and 2 cases a diagnosis of SHGUC and AUC was made, respectively. All the HGUC diagnoses were confirmed by histological biopsies, indicating recurrence of high bladder carcinoma; in seven cases with a diagnosis of SHGUC/AUC, the histology showed high-grade urothelial carcinoma in three cases and, in four cases, a histological diagnosis of epithelial dysplasia was made. In the group of IU without BE results, 32 patients showed a cytological diagnosis of NHGUC, while a diagnosis of HGUC, SHGUC, and AUC was made in five cases, three cases, and two cases, respectively. The histological biopsies confirmed HGUC in eight patients, while in two patients, the biopsy showed flogosis with reactive changes in the urothelial cells.

The second BE test performed showed, in patients with a cytologic diagnosis of NHGUC and AUC, an EpiScore <60; in cases with cytology positive for HGUC, the BE test confirmed the presence of HGUC recurrence, with an EpiScore showing a range from 78 to 95. In the group of patients with a cytological diagnosis of SHGUC, in two cases, the second BE test showed an EpiScore < 60, indicating low risk for high-grade carcinoma.

All cases with an EpiScore < 60 and a cytological diagnosis of NHGUC did not show, during follow-up until now, a clinical or pathological recurrence of high-grade urothelial carcinoma (Table 3).

## 4. Discussion

DNA methylation is an essential epigenetic modification that plays crucial roles in gene regulation, development, and disease and is widely dysregulated in most types of cancer [23]. High-resolution promoter tiling array approaches have been used to analyze DNA methylation in cancer specimens and normal tissue, shedding light on the importance of methylation analysis in cancer research [24]. Furthermore, the Cancer Genome Atlas Pan-Cancer Analysis Project emphasizes the significance of understanding molecular aberrations and their functional roles across tumor types to extend effective therapies from one cancer type to others with a similar genomic profile (Weinstein et al., 2013) [25].

Epigenetic deregulation is a hallmark of cancer, characterized by the frequent acquisition of new DNA methylation in CpG islands, playing an important role in oncogenic pathways involved in cell proliferation [26,27,28].

The Bladder EpiCheck test is a urinary marker based on DNA methylation changes associated with bladder carcinoma in a panel of 15 genomic biomarkers. Numerous studies analyzing the performance of the Bladder EpiCheck test showed high sensitivity and specificity in patients with NMIBC who were under surveillance, allowing us to predict the risk of the neoplastic recurrence of HGUC [2,3], concluding that this test is a robust, high-performing diagnostic test that, as other biomarkers, can potentially reduce the current burden of repeat cystoscopy and cytology tests, improving a patient’s quality of life [8,9,10,29,30,31,32].

The frequency of invalid results in methylation analysis poses a significant challenge, impacting the reliability of diagnostic and prognostic assessments. Several studies have investigated the use of DNA methylation markers in urine samples to accurately predict bladder cancer, identify progression risk, and establish novel methylated genes as urinary tumor markers [33,34,35,36]. These studies have demonstrated the potential of methylation analysis in urine samples for the non-invasive detection and monitoring of bladder cancer.

The reasons for invalid results in methylation analysis can be attributed to various factors. For instance, the analysis of methylation data via certain technologies may remain useful, but the inability to probe methylation at a very high coverage at reasonable costs can lead to invalid results [37]. Moreover, stepwise DNA methylation changes have been linked to escape from defined proliferation barriers and mammary epithelial cell immortalization, indicating the complexity of methylation dynamics and the potential for invalid results in understanding these changes [38]. Furthermore, the development of reasoning with causal conditionals has shown a steady age-related increase in uncertainty responses, which could potentially lead to invalid conclusions in methylation analysis [39].

The location in the genome, consistency, and variation in metachronous tumors, as well as the impact on transcripts and chromosomal location, could identify another factor increasing the number of invalid results in methylation analysis [40]. Additionally, the association of promoter hypermethylation with tumor grade and invasiveness in urothelial bladder cancer highlights the complexity of methylation patterns and their correlation with disease progression [41]. Furthermore, the unique DNA methylation patterns distinguishing non-invasive and invasive urothelial cancers and the establishment of an epigenetic field defect in premalignant tissue underscore the intricate nature of methylation dynamics in bladder cancer [42].

One of the key factors contributing to invalid results is the heterogeneity of DNA methylation patterns in bladder cancer. Studies have shown that the methylation status of specific genes can vary significantly between different tumor stages and grades, leading to challenges in establishing consistent biomarkers for diagnostic and prognostic purposes [40,43,44,45,46,47].

The dynamic nature of DNA methylation in bladder cancer, influenced by environmental factors, aging, and disease progression, further complicates the interpretation of methylation analysis results [40,48].

Additionally, technical limitations and variability in methylation detection methods can contribute to invalid results. The choice of assay, such as bisulfite conversion-based methods or methylation-specific PCR, can impact the accuracy and reproducibility of methylation analysis, potentially leading to inconsistent results. Furthermore, the lack of standardized protocols and quality control measures in methylation analysis can introduce variability and potential sources of error, contributing to the frequency of invalid results [40,43,44,45,46].

Finally, the influence of tumor heterogeneity and the tumor microenvironment on methylation patterns can lead to challenges in interpreting methylation analysis results. The presence of subclonal methylation events and the impact of stromal cells in the tumor microenvironment can introduce complexity and potential sources of variability, contributing to the frequency of invalid results [49,50,51,52,53,54,55,56,57,58,59,60].

In the literature, numerous studies analyzed the way to reduce the frequency of invalid results in methylation analysis, showing that it is essential to consider the type of DNA preferred for mutation analysis, as highlighted in the systematic review on mutation markers for bladder cancer diagnosis in urine [61]. Additionally, the introduction of Infinium DNA Methylation Bead Chip arrays has facilitated the highly reproducible analysis of CpG sites at low cost, offering a potential solution to reduce the frequency of invalid results in large patient cohorts [62]. Moreover, the use of Bayesian hierarchical models and adaptive smoothing methods can improve the power of differential methylation analysis, potentially reducing the occurrence of invalid results [23,63,64].

In the methylation analysis of urinary samples, a number of invalid BE tests is expected in routine practice, and they could be due to multiple reasons such as technical procedural errors or the quality of the sample. The suggestions to fix this in most of these cases is to collect a new urine sample from the patient.

In our study, we analyzed, for the first time to our knowledge, the percentage of invalid BE tests in patients with high-grade urothelial carcinoma during follow-up, investigating if there is a difference in invalid BE results, considering the urine collection method, voided urine (UV) or bladder washing (IU), respectively.

In the group of patients with a BE test performed by collecting UV samples, the percentage of invalid tests was about 14% (58/402), while in the group of samples collected after bladder washing (IU), the value was about 11% (41/380). It is known that reliable results of a BE test are dependent on correct specimen collection and transportation and storage of the urinary sample, and these factors could affect a voided urinary sample (UV) more than a bladder washing sample (IU). Our results seem to indicate that no statistically significant difference was found in the two different groups (*p* = 0.13 Fisher’s exact test), suggesting that several factors were involved in cases with an invalid BE result, which seem to be not related to the collection method of the urinary sample.

Moreover, the second BE test performed showed an EpiScore < 60 in all patients with a previous cytological diagnosis of NHGUC and AUC and an EpiScore > 60 in all patients with previous cytology positive for HGUC or SHGUC, confirmed by histology biopsies.

These data seem to support the hypothesis that, in cases with cytology for NHGUC with a concurrent invalid BE result, a second BE test could be not useful, increasing the economic burden of the follow-up for NMIBC.

In our previous paper analyzing the combined approach of cytology and a methylation test in the follow-up of patients with non-muscle-invasive bladder carcinoma (NMIBC), we demonstrated that, in patients with cytology for NHGUC or HGUC, a BE test confirming the cytological diagnosis could be useless, while only in cases with AUC or SHGUC cytology, the BE test may help to make a correct diagnosis of recurrence of HGUC [9].

Thus, we suggest that, in the follow-up of NMIBC for patients for whom a BE test is planned, a combined approach of cytology and a methylation test is recommended in order to repeat the BE test with an invalid result only in cases with AUC, SHGUC, and HGUC cytological diagnoses.

Our study has some limitations. First, all our data seem to indicate that the collection method of the urinary sample is not involved in an invalid result of BE, showing the versatility of this kind of test that could be used in routine practice in an academic center or private laboratory; but, several factors remain to be investigated. We did not analyze every single error in the EpiCheck run, but we considered all the errors as only one group with the same suggestion: to repeat the BE test with a new urine sample. A more in-depth analysis of the errors could identify a specific factor responsible for most of the invalid results.

Moreover, in our study, in cases with an invalid BE test, we did not repeat the PCR but we collected new urinary samples. It is known that one of the problems in BE could be an incomplete digestion of the sample or an insufficient amount of DNA; we considered that a new specimen would avoid these factors better than repeating the test.

Another possible bias in our paper is that the cytology and the second BE test were from different samples, collected in different moments, even if the clinical data and the follow-up of the patients seemed to reduce these limitations.

Finally, the invalid BE results were laboratory data, linked to a technician’s experience. We tried to avoid this limitation by working with the same technician group after a 1-year-long training.

In conclusion, the methylation analysis of urinary samples in non-muscle-invasive bladder carcinoma holds promise for the non-invasive detection, risk stratification, and monitoring of bladder cancer. However, the frequency of invalid results in the methylation analysis of non-muscle-invasive bladder carcinoma can be attributed to the complex interplay of biological, technical, and environmental factors. Addressing these challenges requires a comprehensive approach, including the standardization of methodologies, consideration of tumor heterogeneity, and the implementation of quality control measures to enhance the reliability and accuracy of methylation analysis in clinical practice.

Further research to identify robust biomarkers and improve the reliability of methylation analysis in clinical practice is needed.

## Figures and Tables

**Table 1 biomedicines-11-03288-t001:** Clinical cohort of 782 patients.

Features	*n* (%)
Sex, *n* (%)	
Male	485 (62)
Female	297 (38)
Age at time of resection	
Range	45–93
Mean	72.5
Diagnosis of HGUC (histology biopsies)	596
G3T1	360
G2T1	236
CIS	186

**Table 2 biomedicines-11-03288-t002:** BE test results in urine voided samples (UV) and instrumented urine samples (IU).

	Valid BE Result	Invalid BE Result	
**UV samples**	**344**	**58**	** *p* ** **=** **0.133**
**IU samples**	**339**	**41**	OR 0.717, 95% CI from 0.467 to 1.1

**Table 3 biomedicines-11-03288-t003:** Cytology, BE test (invalid results and second BE test) in urine voided samples (UV), and instrumented urine samples (IU).

**Cytological Diagnosis**	**Invalid BE UV Result**	**Histological Diagnosis** **Positive for Carcinoma**	**Histological Diagnosis** **Negative for Carcinoma**	**Second BE Episcore < 60**	**Second BE Episcore > 60**
**NHGUC**	**45**	**/**	**/**	**45**	**0**
**AUC**	**2**	**0**	**2**	**2**	**0**
**SHGUC**	**5**	**3**	**2**	**2**	**3**
**HGUC**	**6**	**6**	**0**	**0**	**6**
**Cytological Diagnosis**	**Invalid BE IU Result**	**Histological Diagnosis** **Positive for Carcinoma**	**Histological Diagnosis** **Negative for Carcinoma**	**Second BE Episcore < 60**	**Second BE Episcore > 60**
**NHGUC**	**32**	**/**	**/**	**32**	**0**
**AUC**	**2**	**1**	**1**	**1**	**1**
**SHGUC**	**3**	**2**	**1**	**1**	**2**
**HGUC**	**5**	**5**	**0**	**0**	**5**

## Data Availability

The data presented in this study are available on request from the corresponding author.

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
