# Peer review of "Methylation Analysis of Urinary Sample in Non-Muscle-Invasive Bladder Carcinoma: Frequency and Management of Invalid Result"

_biomedicines, 2023, doi:10.3390/biomedicines11123288_

Round 1

Reviewer 1 Report

Comments and Suggestions for Authors

The investigation aimed to explore the frequency and management of invalid results affecting methylation analysis of urinary samples for detecting bladder carcinoma. The study maintains a clear objective and a coherent flow of content. However, there are a few suggestions for improvement:

  1. Introduction: It would be beneficial to include basic information on non-muscle invasive bladder carcinoma and the standard treatment options.
  2. Introduction, L47-52: Clarification is needed regarding why the study was conducted if the issue of invalid samples can be resolved by retesting with new samples. Further elaboration on this point is essential.
  3. Materials and Methods, Section 2.1: Please specify whether the patient study source obtained board approval. Additionally, an explanation for the greater number of male patients compared to females would be valuable to address potential data bias.
  4. Section 2.3: Please elucidate the rationale behind conducting chi-square and Fisher’s exact tests in this study.
  5. Table 2: Reformatting the table is necessary as the data are not aligned properly in the rows.
  6. Discussion: The authors suggested a combined approach using cytology and methylation tests. It would be insightful if a comparison of results with and without employing the proposed approach could be provided.
  7. It would be beneficial, if feasible, to include a schematic flow chart illustrating the experimental process of this study.
Comments on the Quality of English Language

I have no problem to read this manuscript in English.

Author Response

REVIEWER 1

Comments and Suggestions for Authors

The investigation aimed to explore the frequency and management of invalid results affecting methylation analysis of urinary samples for detecting bladder carcinoma. The study maintains a clear objective and a coherent flow of content. However, there are a few suggestions for improvement:

  1. Introduction: It would be beneficial to include basic information on non-muscle invasive bladder carcinoma and the standard treatment options. 

SPECIFIED  IN THE TEXT  Line 39-49

  1. Introduction, L47-52: Clarification is needed regarding why the study was conducted if the issue of invalid samples can be resolved by retesting with new samples. Further elaboration on this point is essential

SPECIFIED  IN THE TEXT Line 70-72

  1. Materials and Methods, Section 2.1: Please specify whether the patient study source obtained board approval. Additionally, an explanation for the greater number of male patients compared to females would be valuable to address potential data bias.

Answer: The number of male and female are not considered ad potential  data bias in fact the incidence of this kind of neoplasia in male and female is M/F 1:2 or 1:3

  1. Section 2.3: Please elucidate the rationale behind conducting chi-square and Fisher’s exact tests in this study.

The test allows to verify whether the differences in the data may be due to chance; in the event that the test shows that they cannot be the result of chance, we speak of 'statistical significance'. It is used in situations where there are two dichotomous nominal variables and small samples such as in the data of this report.

  1. Table 2: Reformatting the table is necessary as the data are not aligned properly in the rows.

DONE IN THE TEXT

  1. Discussion: The authors suggested a combined approach using cytology and methylation tests. It would be insightful if a comparison of results with and without employing the proposed approach could be provided.

Answer:  The useful of combined approach cytology and methylation test has been demonstrated in our previous paper and confirmed by data present in literature in similar papers.

(Pierconti F, Martini M, Cenci T, Fiorentino V, Gianfrancesco LD, Ragonese M, Bientinesi R, Rossi E, Larocca LM, Racioppi M, Bassi PF. The bladder epicheck test and cytology in the follow-up of patients with non-muscle-invasive high grade bladder carcinoma. Urol Oncol. 2022 Mar;40(3):108.e19-108.e25. doi: 10.1016/j.urolonc.2021.11.013. Epub 2021 Dec 10.PMID: 34903453

Pierconti F, Martini M, Fiorentino V, Cenci T, Capodimonti S, Straccia P, Sacco E, Pugliese D, Cindolo L, Larocca LM, Bassi PF. T The combination cytology/epichek test in non muscle invasive bladder carcinoma follow-up: Effective tool or useless expence? Urol Oncol. 2021 Feb;39(2):131.e17-131.e21. doi: 10.1016/j.urolonc.2020.06.018. Epub 2020 Aug 7.PMID: 32773233

Trenti E, D'Elia C, Mian C, Schwienbacher C, Hanspeter E, Pycha A, et al. Diagnostic predictive value of the Bladder EpiCheck test in the follow-up of patients with non-muscle-invasive bladder cancer. Cancer Cytopathol. 2019;127:465-469. doi: 10.1002/cncy.22152. )

Reviewer 2 Report

Comments and Suggestions for Authors

In this study, seven hundred-eighty-two patients with diagnosis of non-muscle invasive high grade carcinoma. The Bladder EpiCheck test was performed together with cytology in all cases, within 1 year after the end of treatment. In 402 patients the urinary samples were voided urine while in 380 cases the samples were collected after bladder washing. For all the patients with invalid BE results, a second BE test was performed following the Instructions for use that indicate the test should be repeated with a new urinary sample in case of invalid result. By analyzing the two different groups, authors found the BE invalid results seems to be not related to urinary samples (p=0,13 Fisher’s exact test) suggesting that the collection method is not relevant in order to reduce the number of invalid test. For the patients for which a BE test is planned, a combined approach cytology and methylation test is recommended, in order to repeat the BE test with invalid result only in cases with cytological diagnosis of Atypical Urothelial Cells (AUC), Suspicious for High Grade Urothelial Carcinoma (SHGUC) and High Grade Urothelial Carcinoma (HGUC). The study is comprehensive and the following issues should be addressed. A big concern of this study is that authors should perform the statistical analysis using the one-way ANOVA and to show the results as * p < 0.05, ** p < 0.01, and *** p < 0.001, et al.

Author Response

REVIEWER 2

Comments and Suggestions for Authors

In this study, seven hundred-eighty-two patients with diagnosis of non-muscle invasive high grade carcinoma. The Bladder EpiCheck test was performed together with cytology in all cases, within 1 year after the end of treatment. In 402 patients the urinary samples were voided urine while in 380 cases the samples were collected after bladder washing. For all the patients with invalid BE results, a second BE test was performed following the Instructions for use that indicate the test should be repeated with a new urinary sample in case of invalid result. By analyzing the two different groups, authors found the BE invalid results seems to be not related to urinary samples (p=0,13 Fisher’s exact test) suggesting that the collection method is not relevant in order to reduce the number of invalid test. For the patients for which a BE test is planned, a combined approach cytology and methylation test is recommended, in order to repeat the BE test with invalid result only in cases with cytological diagnosis of Atypical Urothelial Cells (AUC), Suspicious for High Grade Urothelial Carcinoma (SHGUC) and High Grade Urothelial Carcinoma (HGUC). The study is comprehensive and the following issues should be addressed.

A big concern of this study is that authors should perform the statistical analysis using the one-way ANOVA and to show the results as * p < 0.05, ** p < 0.01, and *** p < 0.001, et al.

Answer: The Fisher’s exact test allows to verify whether the differences in the data may be due to chance; in the event that the test shows that they cannot be the result of chance, we speak of 'statistical significance'. It is used in situations where there are two dichotomous nominal variables and small samples such as in the data of this report. One-way ANOVA is a statistical technique created in the field of experimental research to assess the effect of certain factors, independent variables - continuous or categorial type, on the dependent variable - continuous type. But in our case we have a nominal variables and not continuous variables.

Reviewer 3 Report

Comments and Suggestions for Authors

This study analyzed a cohort of 782 patients with urothelial Non-muscle-
invasive Bladder Cancer (NMIBC), and carcinoma in situ (CIS).

The diagnosis of high-risk urothelial cancer is necessary as their clinical behavior is aggressive and likely to progress to invasive stage.

This research performed the Bladder EpiCheck test in cytology samples in all cases.

The Bladder EpiCheck test is an in vitro diagnostic device for the detection of DNA methylation patterns in urine that are associated with urothelial carcinoma. It is intended for use as a noninvasive method for monitoring of tumor recurrence in conjunction with standard diagnostic procedures in patients previously diagnosed with bladder cancer and/or upper tract urothelial carcinoma. The test analyzes disease-specific changes in DNA methylation markers, allowing for the detection of high-risk (non Ta-LG) cancers.

Comments:

(1) Could you please describe the sensitivity, specificity, and negative/positive Predictive Value (NPV) of this test?

(2) Line 65, was the initial diagnosis made using diagnostic biopsies fixed in formalin and paraffin-embedded (FFPE) tissue samples?

(3) Line 68, regarding according to the WHO. Could you please specify the year of the WHO classification? Is it the latest classification?

(4) In Table 1. As I understand, based on the 2009  TNM classification, updated in 2017 (8th Edn.), all cases were Tis (Carcinoma in situ: ‘flat tumour’) or T1 (Tumour invades subepithelial connective tissue). Is this correct?

(5) Please define the histological and cytological criteria for HGUC.

(6) Regarding the clinicopathological characteristics of the cohort. Only sex, age, histological diagnosis is provided. Are other variables available? Or this study was restricted to 3 variables? What about risk factors, treatment (line 81-->BCG and mitomycin-c only? line 92 --> resection), response to treatment, prognostic factors, etc.?

(7) Lines 91 - 105. It may be my mistake, but I am not sure when the cytological analysis was performed. Is it within 1 year after the original treatment? Therefore, it can range from 1 day to 365 days? In some cases, the cytological diagnosis was confirmed by biopsy, is this right?

(8) Line 112. The cytological diagnosis was NHGUC, AUC, SHGUC, HGUC, and US/ND. Is this correct?

(9) Line 124. The epicheck was <60 negative (low-risk), 60-90 positive high-risk, and 90-100 positive diagnosis HGUC.  Is the low-grade UC always included in the <60 range?

(10) Could you please provide the details of the epicheck test? What genes were analyzed? How the software works? Is it a "source closed" method?

(11) In Table 2, is the OR calculation necessary?

(12) Talbe 3 shows how a 2nd epicheck improves the sensitivity. This table focuses on the analysis of invalid results.  But, is it possible to make a crosstabulation as:

rows (cytology): NHGUC, AUC, SHGUC, HGUC, and US/ND

columns (epicheck): low-risk, high-risk, HGUC

using the final epicheck results. In case of invalid, please use the valid.

Why the analysis of valid cases is not shown? I am aware that the title is focused on invalid cases, but it is worth knowing if epicheck is valid method for relapse diagnosis.

Author Response

REVIEWER 3

This study analyzed a cohort of 782 patients with urothelial Non-muscle-
invasive Bladder Cancer (NMIBC), and carcinoma in situ (CIS).

The diagnosis of high-risk urothelial cancer is necessary as their clinical behavior is aggressive and likely to progress to invasive stage.

This research performed the Bladder EpiCheck test in cytology samples in all cases.

The Bladder EpiCheck test is an in vitro diagnostic device for the detection of DNA methylation patterns in urine that are associated with urothelial carcinoma. It is intended for use as a noninvasive method for monitoring of tumor recurrence in conjunction with standard diagnostic procedures in patients previously diagnosed with bladder cancer and/or upper tract urothelial carcinoma. The test analyzes disease-specific changes in DNA methylation markers, allowing for the detection of high-risk (non Ta-LG) cancers.

Comments:

(1) Could you please describe the sensitivity, specificity, and negative/positive Predictive Value (NPV) of this test?

SPECIFIED  IN THE TEXT Line 57-58

 (2) Line 65, was the initial diagnosis made using diagnostic biopsies fixed in formalin and paraffin-embedded (FFPE) tissue samples?

YES

(3) Line 68, regarding according to the WHO. Could you please specify the year of the WHO classification? Is it the latest classification? WHO 1973.

SPECIFIED  IN THE TEXT Line 80-83

(4) In Table 1. As I understand, based on the 2009  TNM classification, updated in 2017 (8th Edn.), all cases were Tis (Carcinoma in situ: ‘flat tumour’) or T1 (Tumour invades subepithelial connective tissue). Is this correct?

YES

 (5) Please define the histological and cytological criteria for HGUC.

SPECIFIED  IN THE TEXT Line 137-141; line 111-114

(6) Regarding the clinicopathological characteristics of the cohort. Only sex, age, histological diagnosis is provided. Are other variables available? Or this study was restricted to 3 variables? What about risk factors, treatment (line 81-->BCG and mitomycin-c only? line 92 --> resection), response to treatment, prognostic factors, etc.?

Answer: Familiarity of neoplasia or smoking are variables not available for this patients. Anyway the variables considered are the most common variables considered in all papers which investigated on this kind of tumor

(7) Lines 91 - 105. It may be my mistake, but I am not sure when the cytological analysis was performed. Is it within 1 year after the original treatment? Therefore, it can range from 1 day to 365 days? In some cases, the cytological diagnosis was confirmed by biopsy, is this right?

Answer: The cytological analysis was performed within 1 year after the treatment at 3-6 months, following the guideline for this tumor. The cytological diagnosis of HGUC or SHGUC were confirmed by biopsies

(8) Line 112. The cytological diagnosis was NHGUC, AUC, SHGUC, HGUC, and US/ND. Is this correct?

YES. These category are indicated by PARIS SYSTEM for reporting urinary samples

(9) Line 124. The epicheck was <60 negative (low-risk), 60-90 positive high-risk, and 90-100 positive diagnosis HGUC.  Is the low-grade UC always included in the <60 range?

Answer: The Bladder Epicheck indicated low risk and high risk for HGUC. The low grade UC are generally included in the BE low risk group

(10) Could you please provide the details of the epicheck test? What genes were analyzed? How the software works? Is it a "source closed" method?

Answer: The BE test analyze 15 genes involve in bladder carcinogenesis of High grade urothelial carcinoma, the name of this genes are not indicated by the company that produce the test  

(11) In Table 2, is the OR calculation necessary?

Answer: Yes. In fact, the OR is a statistic that measures the degree of correlation between two factors, that in our case is moderate (near 0.6).

(12) Talbe 3 shows how a 2nd epicheck improves the sensitivity. This table focuses on the analysis of invalid results.  But, is it possible to make a crosstabulation as:

rows (cytology): NHGUC, AUC, SHGUC, HGUC, and US/ND

columns (epicheck): low-risk, high-risk, HGUC

using the final epicheck results. In case of invalid, please use the valid.

Why the analysis of valid cases is not shown? I am aware that the title is focused on invalid cases, but it is worth knowing if epicheck is valid method for relapse diagnosis.

The table and the goal of this paper is the evaluation of invalid test. We have demonstrated in numerous paper that this test is a valid method for relapse of diagnosis of bladder carcinoma (Pierconti F, Martini M, Cenci T, Fiorentino V, Gianfrancesco LD, Ragonese M, Bientinesi R, Rossi E, Larocca LM, Racioppi M, Bassi PF. The bladder epicheck test and cytology in the follow-up of patients with non-muscle-invasive high grade bladder carcinoma. Urol Oncol. 2022 Mar;40(3):108.e19-108.e25. doi: 10.1016/j.urolonc.2021.11.013. Epub 2021 Dec 10.PMID: 34903453

Pierconti F, Martini M, Fiorentino V, Cenci T, Capodimonti S, Straccia P, Sacco E, Pugliese D, Cindolo L, Larocca LM, Bassi PF. T The combination cytology/epichek test in non muscle invasive bladder carcinoma follow-up: Effective tool or useless expence? Urol Oncol. 2021 Feb;39(2):131.e17-131.e21. doi: 10.1016/j.urolonc.2020.06.018. Epub 2020 Aug 7.PMID: 32773233

Round 2

Reviewer 1 Report

Comments and Suggestions for Authors

I am satisfied with the modifications and explanation of the authors as per my comments in the revision. The quality and presentation of this work are improved.

Comments on the Quality of English Language

I do not have problem to read this revised manuscript.